# A Survey of Paediatric Rapid Sequence Induction in a Department of Anaesthesia

**DOI:** 10.3390/children9091416

**Published:** 2022-09-19

**Authors:** Lloyd Duncan, Michelle Correia, Palesa Mogane

**Affiliations:** 1Department of Anaesthesiology, School of Clinical Medicine, Faculty of Health Sciences, University of the Witwatersrand, Johannesburg 2000, South Africa; 2Department of Anaesthesiology, Nelson Mandela Children’s Hospital, Johannesburg 2000, South Africa

**Keywords:** paediatric, rapid sequence induction, anaesthesia

## Abstract

(1) Background: Rapid sequence induction (RSI) is carried out by anaesthetists to secure the airway promptly in patients who are at risk of aspirating gastric content during induction of anaesthesia. RSI requires variation in the paediatric population. We conducted a survey to investigate current practice of paediatric RSI by anaesthetists. (2) Methods: A descriptive, contextual, cross-sectional research design was followed. The study population consisted of all anaesthetists working in the Department of Anaesthesia at the University of the Witwatersrand. Data was collected in the form of a self-administered questionnaire. (3) Results: Of 138 questionnaires that were distributed, 126 were completed. Clinical indication for RSI was predominantly for appendicitis with peritonitis (115/124; 92.7%). Preoxygenation was performed by 95.1% of anaesthetists for children, 87% for infants and 89.4% for neonates. Cricoid pressure was used significantly more in children (56%) than in infants (20.8%) and neonates (10.3%) (*p* < 0.001). Rocuronium was the paralytic agent of choice in children (42.7%) and infants (38.2%), while cisatracurium was used most frequently in neonates (37.4%). Suxamethonium was used least in neonates. Cuffed ETTs were used most frequently for children (99.2%) and least for neonates (49.6%). Eighty-five percent of anaesthetists omitted cricoid pressure during RSI for pyloromyotomy, for which a controlled RSI was performed more by consultants and senior registrars (*p* < 0.01). A classic RSI was performed by 53.6% of anaesthetists for laparotomy for small bowel obstruction. Consultants and PMOs were more likely to intubate a child for forearm MUA who was starved for 6 h and received opioids (*p* < 0.05). Controlled RSI with cisatracurium was the technique of choice for Tenkhoff insertion in a child with renal failure. (4) Conclusions: RSI practice for paediatric patients varied widely among anaesthetists. This may be attributed to a combination of anaesthetic experience, training in paediatric anaesthesia, and patient specific factors, along with the individualised clinical scenario’s aspiration risk. A controlled RSI technique appears to be implemented more frequently by anaesthetists with increased experience.

## 1. Introduction

Rapid sequence induction (RSI) was introduced into anaesthetic practice to prevent aspiration of gastric content into the lungs [1]. The steps of a classic RSI do not take into account the physiological, anatomical and psychological differences in neonates, infants and children [2]. Technique variation, described as controlled RSI, balances the risk of aspiration with the more prevalent risk of hypoxaemia during classic RSI [3]. The conduct of RSI in paediatric patients is performed with marked variation and appears to differ in relation to the practitioner’s experience in paediatric anaesthesia [4].

Classic RSI is fundamental to aspiration risk mitigation. Its key features include pre-oxygenation, intravenous induction, application of cricoid pressure, administration of suxamethonium and a period of apnoea until paralysis allows endotracheal intubation [1]. The direct application of this to the paediatric patient, however, may be stressful and harmful [5]. Pulmonary aspiration with subsequent poor patient outcome during tracheal intubation is a much-feared complication of anaesthesia, but it is rare in the paediatric population (2.2 per 10,000 emergency cases) with no mortality reported [6].

A controlled RSI technique addresses aspiration risk while tailoring the induction to paediatric physiological requirements. Its key features include appropriate bag-mask ventilation to maintain oxygenation and adequate depth of anaesthesia and paralysis prior to intubation [3]. Although the incidence of pulmonary aspiration is extremely low in patients receiving controlled and classic RSI, patients managed with a classic RSI technique have been found to have significantly higher episodes of desaturation, bradycardia and difficult intubation [7].

A study conducted in England in 2007 surveyed the RSI practice of 369 anaesthetists with varying degrees of experience in paediatric anaesthesia [4]. Results showed that technique varied according to patient age and level of experience of the anaesthetist. Significant findings included that cricoid pressure was used less often on infants than on school children, and that suxamethonium was widely used in both infants and older children but was less likely to be used by consultants than trainee anaesthetists.

Published data on paediatric RSI practice in our academic department setting is lacking. There are currently no guidelines or specific training on RSI practice within the department. The aims of the study were to investigate the conduct of RSI in paediatric patients by anaesthetists at the University of the Witwatersrand and to determine whether practice varies in relation to paediatric training, level of experience, age group of the patient and the clinical scenario for which RSI is indicated.

## 2. Materials and Methods

A descriptive, contextual, cross-sectional study was conducted which employed a self-administered questionnaire to the entire accessible population of medical officers (MO), registrars, principal medical officers (PMO) and consultants. An MO is a qualified doctor working in the department under specialist supervision. A PMO is a doctor with more than 10 years of anaesthetic experience and is regarded as a consultant. A convenience sampling method was used. 

We calculated the study sample size in consultation with a biostatistician who used Raosoft^®^ sample size calculator. Using our department population size of 206 anaesthetists, the recommended sample size was calculated as 124. This is in keeping with the minimum response distribution of 60% considered acceptable for a questionnaire [8].

Permission to use and adapt a previously published questionnaire by Stedeford and Stoddart [4] was obtained. Further construction of the questionnaire was based on an unpublished survey performed by one of the authors of this study. To achieve further content and face validity, three consultant anaesthetists with an interest in the field of study were consulted to peer-review the questionnaire.

The questionnaire consisted of three sections (see Appendix A). The first section included questions that ascertained level of training and paediatric experience of the anaesthetist as well as two RSI knowledge-based questions. The second section contained 10 practice-based questions which focused on what the anaesthetist’s standard RSI technique would be in neonates, infants and children. The final section consisted of four different clinical scenarios in which the anaesthetists were asked to indicate what their own stepwise conduct of the RSI would be for each given scenario. The scenarios evaluated the conduct of anaesthesia for:A 3-week-old, otherwise healthy baby, for a pyloromyotomy that had been fully resuscitated with intravenous (IV) fluids, but the cannula came out during transfer to theatre. The patient had a nasogastric tube in situ that appeared to be draining well.A 4-year-old, previously healthy child for exploratory laparotomy who had suspected small bowel obstruction. The patient had been unwell for 48 h, with episodes of vomiting and a tender, distended abdomen. The patient had an IV line running and had been resuscitated, but there was no nasogastric (NG) tube in place.A 6-year-old, otherwise healthy child who had a painful forearm fracture requiring manipulation under anaesthesia (MUA). The patient had eaten two hours prior to the injury but had been starved for six hours since the injury and had received opioids in the pre-operative period.A 7-year-old with renal failure and ascites who had been booked for Tenckhoff catheter insertion for peritoneal dialysis. It was an elective procedure, and the child had been starved.

For each scenario, the anaesthetist was instructed to describe the technique they would most commonly use to establish anaesthesia by indicating on the table the components they would use and the order in which they would use them (Table 1). Techniques were also assessed for features consistent with classic or controlled RSI.

Data were collected by distribution of the questionnaires at departmental academic meetings from February to August 2020. Those who agreed to participate were given an information letter and the questionnaire. One researcher remained at the meeting to be available to answer any questions and to prevent data contamination. The completed questionnaires were placed into a sealed box. Each questionnaire was assigned a number. Questionnaires that were returned blank were also assigned a number and included for response rate calculation but not for data interpretation.

Statistical analysis was conducted in consultation with a statistician. Frequencies and percentages were used to describe categorical variables. Pearson’s Chi-squared tests and Fisher’s exact tests (where data were sparse) were used to compare differences in responses by anaesthetist grade or patient age group. Bar graphs were used to describe RSI indication, controlled RSI components, pre-oxygenation, cricoid pressure, suxamethonium use, and choice of airway by anaesthetist’s grade. Data capturing and bar graphs were performed in Microsoft Excel^®^ 2016, and statistical analysis was performed in Stata version 15^®^ (StataCorp, College Station, TX, USA). A *p*-value of <0.05 was considered statistically significant.

## 3. Results

Of 138 questionnaires that were distributed, 126 were completed (Table 2), giving a response rate of 91.3%, and a response distribution of 61.1%.

The results of the knowledge and practice-based sections are presented below. Not all questions were answered by all anaesthetists; thus, the number (*N*), or denominator indicated, reflects the completed responses for that particular question.

Figure 1 displays the indications for RSI (classic or controlled) by grade of anaesthetist (*N* = 124). MOs and senior registrars were significantly more likely to perform an RSI for ‘lower GIT obstruction’ than other grades of anaesthetist *(p* < 0.01).

The results for anaesthetists’ choice on their 2 most important components of a controlled RSI are presented in Figure 2. There was no significant difference in responses on the most important components by grade of anaesthetist (*p* = 0.114).

### 3.1. Results for Standard RSI Technique

Anaesthetists were asked to indicate their RSI technique in neonates, infants and children. Results for preoxygenation are shown in Figure 3. The tendency to preoxygenate did not differ significantly by age group (*p* = 0.110). Children were preoxygenated using normal tidal breaths by 76.9% (*n/N* = 90/117) of the anaesthetists of which 80% did this for a duration of 3 min or more and 20% for less than 3 min. Vital capacity breaths were used by 23.1% (*n/N* = 27/117) of which 76% used five to 10 breaths and 24% used less than five breaths. There was no significant difference in grade of anaesthetist and preoxygenation technique for any age group (*p* = 0.221).

Propofol was the most frequently used induction agent across all age groups. For children, infants, and neonates it was used by 78% (*n/N* = 96/123), 67.2% (*n/N* = 82/122) and 62% (*n/N* = 75/121) of anaesthetists, respectively. Sevoflurane was the next favoured induction agent and was used by 33.1% (*n/N* = 40/121) of anaesthetists in neonates, 30.3% (*n/N* = 37/122) in infants and 20.3% (*n/N* = 25/123) in children. Etomidate and ketamine were very seldom used (less than 4% of anaesthetists in all age groups).

Cricoid pressure was used significantly more in children (70/125; 56%) than in infants (26/125; 20.8%) and neonates (13/126; 10.3%) (*p* < 0.001) by all grades of anaesthetist, as shown in Figure 4.

Rocuronium was the anaesthetists’ muscle relaxant of choice for children and infants (53/124; 42.7% and 47/123; 38.2%, respectively), while cisatracurium was the most frequently chosen agent for neonates (46/123; 37.4%). Consultants used cisatracurium more frequently in neonates than PMOs, registrars, and MOs (55.6% vs. 0%, 40% and 9.1%, respectively). Respondents preferentially indicated that they did not use muscle relaxants in neonates (39/123; 31.7% vs. 26/123; 21.1% in infants and 13/124; 10.3% in children), however this technique was employed least by consultants than by any other grade of anaesthetist.

When suxamethonium was used, it was more frequently used in children (38/124; 30.6% vs. 19/123; 15.4% in infants and 10/123; 8.1% in neonates). However, when suxamethonium was used in neonates, consultants were significantly more likely to use it than any other grade of anaesthetist (*p* < 0.05). The results for the use of suxamethonium are presented in Figure 5.

Bag-mask ventilation (BMV) was used by anaesthetists during RSI most frequently for neonates (77/126; 61.1%), followed by infants (59/126; 46.8%) and least in children (32/125; 26.6%). Consultants and registrars were significantly more likely to BMV neonates and infants than PMOs and MOs (*p* < 0.05). Of the anaesthetists that would BMV during RSI (*n* =100), the majority (66%) would do so at an inspiratory pressure of less than 12 cmH20, followed by 33% at 12–16 cmH20 and 1% at more than 16 cmH20. Consultants and registrars performed BMV at an inspiratory pressure of less than 12 cmH20 more frequently than PMOs and MOs (59.4% and 57.4% vs. 20% and 34.8%, respectively).

Nerve stimulator monitoring of adequate paralysis prior to intubation was used by 1.6% (*n/N* = 2/124) of anaesthetists. Ultrasound evaluation of gastric content was used by 0.8% (*n/N* = 1/126). The evaluation of volume status of the patient prior to RSI was performed by 78.6% (*n/N* = 99/126). Most anaesthetists (117/126; 92.9%) reported they would find it useful to have guidelines detailing the conduct of RSI in paediatric patients. 

For RSI in neonates, uncuffed endotracheal tubes (ETT) were marginally favoured over cuffed ETTs (63/125; 50.4% vs. 62/125; 49.6%). Respondents indicated that they were more likely to use a cuffed ETT in older children (112/126; 88.9% in infants and 125/126; 99.2% in children). Selection of cuffed vs. uncuffed ETT varied little between grade of anaesthetist for any age group (*p* = 0.1).

#### 3.1.1. Scenario 1: A 3-Week-Old Neonate for Pyloromyotomy

The 3-week-old neonate for pyloromyotomy was intubated by all but one anaesthetist (120/121; 99.2%), who opted for the use of a laryngeal mask airway (LMA). An uncuffed ETT was chosen by 28.1% (*n/N* = 34/121). Suctioning on the NG tube prior to intubation was performed by 89.9% *(n/N* = 107/119) of anaesthetists. Preoxygenation was carried out by 91% (*n/N* = 112/123). Cricoid pressure was omitted by the majority (102/120; 85%) of anaesthetists and this was independent of grade. Volatile induction was opted for by 41.5% (*n/N* = 49/118) and the remainder chose an IV induction (69/118; 58.5%). Cisatracurium was the commonest muscle relaxant used (46/93; 49.5%). BMV during RSI was performed by 66.9% (*n/N* = 79/118) of anaesthetists. Less than a third of anaesthetists elected to intubate without paralysis (26/119; 28%), of which MOs formed the majority (11/26; 42.3%). A controlled RSI technique was used by 62.7% (*n/N* = 74/118) of anaesthetists and was carried out significantly more by consultants and senior registrars than other grades of anaesthetist (*p* < 0.01).

#### 3.1.2. Scenario 2: A 4-Year-Old with Suspected Small Bowel Obstruction for Laparotomy

A classic RSI technique was used by 53.6% (*n/N* = 67/125) of anaesthetists. NG tube insertion prior to induction was performed by 77% (*n/N* = 97/126) of anaesthetists. Cricoid pressure was applied by 56.4% (*n/N* = 71/126). The majority of respondents administered an IV induction agent (117/124; 94.4%) while the remaining induced with volatile (7/117; 5.6%). Paralysis with rocuronium was used most frequently (54/124; 43.6%), followed by suxamethonium (44/124; 35.5%). Most respondents (82/126; 65.1%) did not BMV during induction. Endotracheal intubation was performed by almost all anaesthetists (123/124; 99.2%). Results are shown in Table 3.

#### 3.1.3. Scenario 3: A 6-Year-Old Booked for Forearm Fracture MUA Who Was Starved for Six Hours and Received Opioids

Regarding airway management, 61.3% (*n/N* = 76/124) of anaesthetists would use an LMA. Registrars and MOs used an LMA more frequently than a cuffed ETT, as opposed to PMOs and consultants, whose choice was more frequently a cuffed ETT than an LMA (*p* < 0.05) (Figure 6). Of those who used an ETT, 18.4% (*n/N* = 9/49) performed a classic RSI. The majority of anaesthetists preoxygenated (113/126; 89.7%), induced using volatile (106/126; 84.1%) and did not use cricoid pressure (106/126; 84.1%).

#### 3.1.4. Scenario 4: A 7-Year-Old with Renal Failure and Ascites for Elective Tenkhoff Catheter Insertion

Intubation was the definitive airway of choice by 96.8% (*n/N* = 120/124) of anaesthetists, of whom 65% (*n/N* = 78/120) employed a controlled RSI technique. Although the choice of induction method did not differ by much (52.4% IV agent vs. 47.6% volatile), MOs and PMOs were more likely to perform an IV induction than registrars and consultants (*p* < 0.01). Most anaesthetists (119/126; 94%) pre-oxygenated the patient and 73% (*n/N* = 92/126) omitted cricoid pressure. Cisatracurium was used by 69.4% (*n/N* = 86/124) of anaesthetists and was the most popular paralysing agent among all grades.

## 4. Discussion

Rapid sequence induction in paediatrics is a controversial issue, and there is a great variety in practice among practitioners [9]. Consistency in aspiration risk assessment was demonstrated in this study, with the most frequent selection of clinical indication for RSI being appendicitis with distended, peritonitic abdomen.

Anaesthetists showed a preference for RSI in cases of bowel obstruction or ileus with a high aspiration risk. This practice is in keeping with the recommendations made by Warner et al. [10] in a study evaluating perioperative pulmonary aspiration, in which the majority of infants and children who aspirated indeed had bowel obstruction or ileus, especially those younger than 3 years of age. Warner et al. [10] also provided evidence that aspiration mainly occurs when there is coughing and straining due to insufficient anaesthesia at induction and tracheal intubation.

In order to prevent this, a controlled RSI has subsequently emerged as a safer technique. This technique aims to: (1) prevent hypoxemia by gentle ventilation; and (2) prevent aspiration by intubating only when ideal conditions are observed, which include demonstrated complete muscle paralysis and a deep level of anaesthesia [7]. The results of this study were in-line with these recommendations as the majority of anaesthetists, independent of grade, indicated a preference for these two practices when asked what components of a controlled RSI were the most important.

In the controlled RSI guidelines for the Children’s Hospital in Zurich, Switzerland, Neuhaus et al. [3] recommend the use of preoxygenation by face mask with 100% oxygen for 2 to 3 min if possible. Time to allow adequate denitrogenation may not always be achievable in the non-compliant or combative child and may actually result in increased oxygen consumption [2,7]. This was reflected in the survey conducted by Stedeford and Stoddart in 2007 in England where anaesthetists were significantly less likely to preoxygenate infants than schoolchildren [4]. In this study, the tendency to preoxygenate was high for neonates, infants, and children. This may be explained by the fact that if optimal preoxygenation can be achieved for 2 min, then apnoeic time to desaturation can be extended from 25 to 60 s [11]. The recommendation is to use preoxygenation as well as soft ventilation after induction so that apnoea with potential hypoxaemia can be avoided altogether [3].

The efficacy of cricoid pressure may be limited in paediatric patients. In children less than 8 years old the anatomy is such that the alignment of the upper esophagus differs between trachea and cervical vertebrae [12]; cricoid pressure may distort visualization of the airway, resulting in a more difficult intubation; and lastly, untimely application of cricoid pressure may result in bucking, straining, regurgitation and aspiration [13]. Cricoid pressure was used by only 56% of anaesthetists in children in our survey, and significantly less in infants and neonates (*p* < 0.001). This differs from the survey conducted by Stedeford and Stoddart [4] in which 96% of anaesthetists would apply cricoid pressure to children in an RSI, but may be ascribed to the categories in the study being only infant and ‘schoolchild’. The guidelines developed by Neuhaus et al. [3] emphasize no cricoid pressure unless in cases of alchalasia, Zenker’s diverticulum, or colon interposition for oesophageal replacement.

Rocuronium was the muscle relaxant of choice for RSI in children and infants in this study. This differs from the survey conducted by Stedeford and Stoddart [4] in which suxamethonium was widely used. This may be explained by an increased awareness for its potential adverse effects in paediatric patients [14], and the demonstration that, at an appropriate dose, rocuronium can create the same ideal intubating conditions as suxamethonium within one minute [15]. When suxamethonium was used by respondents in this study, it was mostly used by consultants providing anaesthesia for neonates. However, suxamethonium remains the first choice drug for RSI in paediatrics, according to 2019 French guidelines, if no contra-indications exist [16].

The use of gastric ultrasound (GUS) to accurately assess presence of gastric content before induction is well described in the literature [17], but was used infrequently in this study. GUS requires training and many supervised scans for the inexperienced practitioner to achieve 95% accuracy [18]. Again, the lack of use of nerve stimulators in this study to establish adequate paralysis during RSI is likely to be explained by their lack of availability and, in turn, knowledge on their use.

Cuffed ETTs were chosen more frequently in our survey for the intubation of infants and children, while uncuffed ETTs were marginally more favoured for use in neonates. This may be attributed to the long term ideology that the airway of the neonate and infant was funnel-shaped (or conical) with its narrowest point at the cricoid. This has resulted in the use of uncuffed ETTs being common practice for this age group [19]. However, more recent studies have shown that there is no significant change in the anterior-posterior to transverse ratio of the airway with age, and that the airway is in fact elliptical in shape and not circular [20]. This new understanding along with improved cuff technology has resulted in the increasing use of cuffed ETTs in neonates and infants [21]. Uncuffed ETTs may still be the ETT of choice in premature neonates or those less than 3 kg as appropriately sized cuffed ETTs are currently not available [19].

In this study, for the neonate coming for pyloromyotomy, 89.9% of anaesthetists chose to have suction applied to the NG tube to evacuate gastric content prior to induction. This is in keeping with the recommendation from a recent revalidation article which encourages aspirating the NG tube in the supine, left lateral decubitus, prone and right lateral decubitus positions [22]. Although only chosen by 41.5% of anaesthetists in our survey, volatile induction has emerged as a safe induction technique for pyloromyotomy [23]. This may indicate a lack in awareness among anaesthetists that this is a safe alternative and continued medical education may help to increase this awareness.

Despite the merits of controlled over classic RSI having been discussed in this article, classic RSI seemed to be marginally favoured (53.6%) as the technique of choice for Scenario 2. It would seem that for cases with undoubtedly high aspiration risk, classic RSI, in which ventilation with potential gastric insufflation and regurgitation is avoided, is used more readily by participants.

## 5. Conclusions

RSI practice for paediatric patients varied widely among anaesthetists. This may be attributed to a combination of anaesthetic experience, training in paediatric anaesthesia, and patient specific factors, along with the individualised clinical scenario’s aspiration risk. A controlled RSI technique appears to be implemented more frequently by anaesthetists with increased practice in paediatrics, and suggests experiential support of the technique. Specific training and awareness of this intubation technique with a departmental protocol may aid in reducing practice variation during paediatric RSI among anaesthetists with differing paediatric experience.

## Figures and Tables

**Figure 1 children-09-01416-f001:**
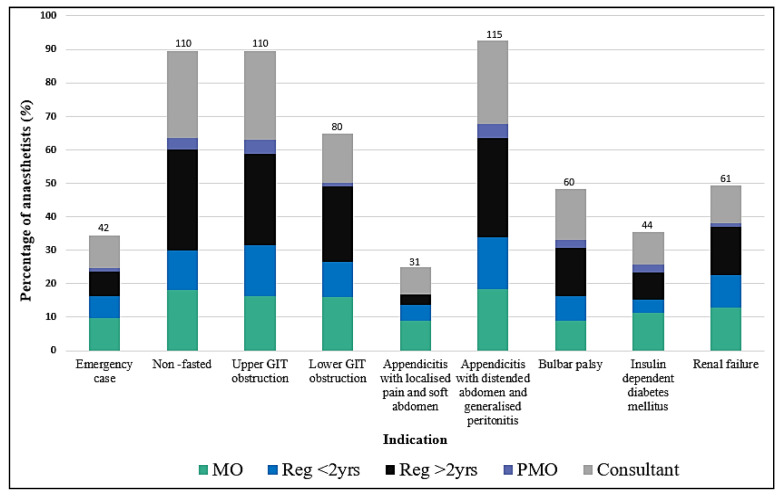
Indications for RSI technique by grade of anaesthetist.

**Figure 2 children-09-01416-f002:**
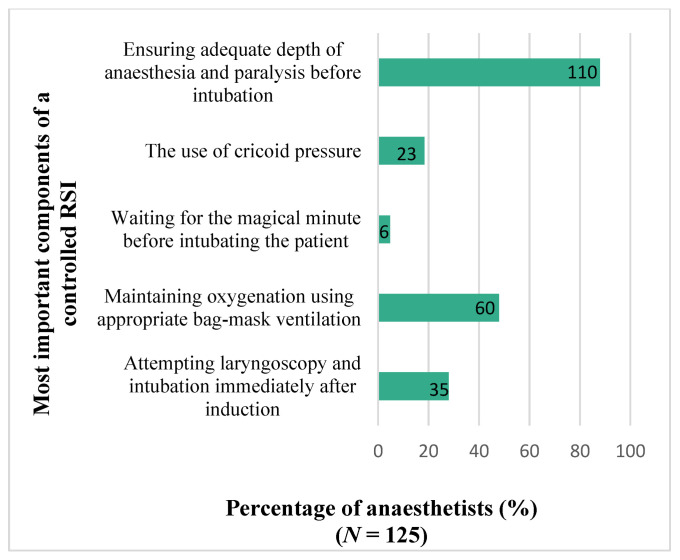
Choice on most important RSI components.

**Figure 3 children-09-01416-f003:**
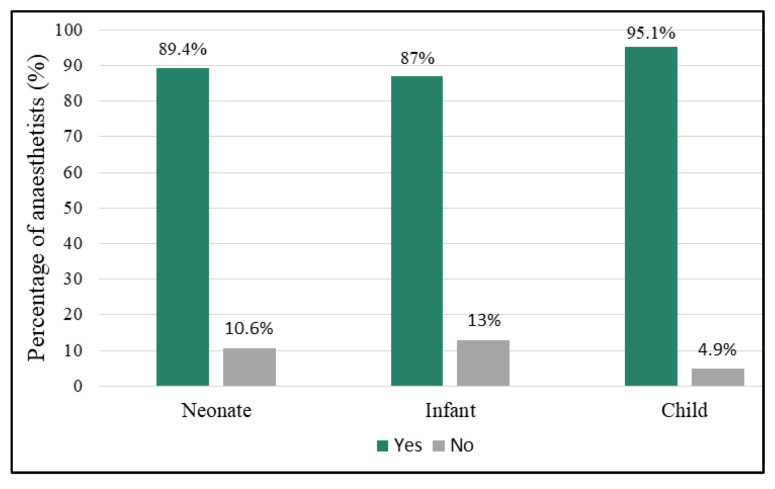
Preoxygenation during RSI.

**Figure 4 children-09-01416-f004:**
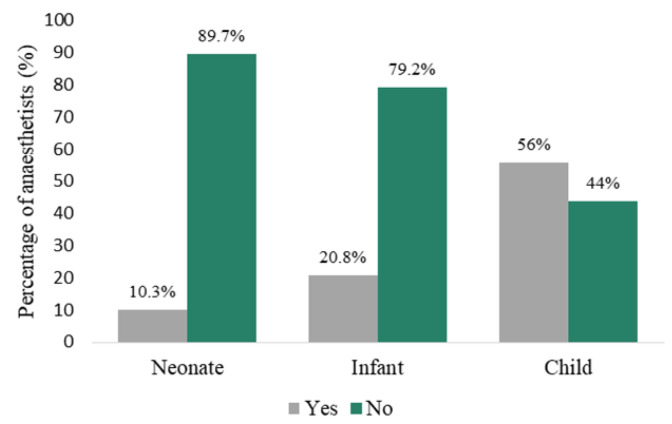
Use of cricoid pressure during RSI.

**Figure 5 children-09-01416-f005:**
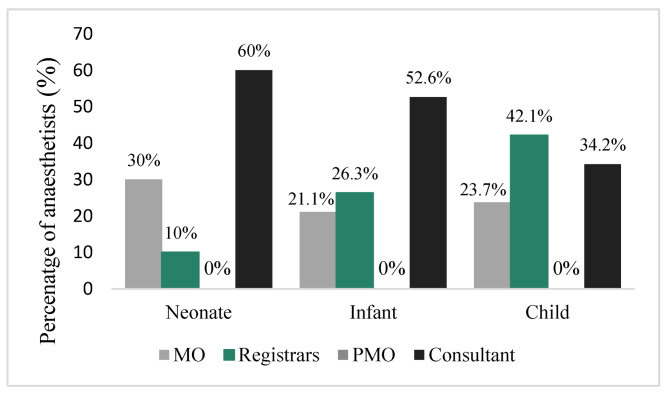
Use of suxamethonium during RSI by grade of anaesthetist.

**Figure 6 children-09-01416-f006:**
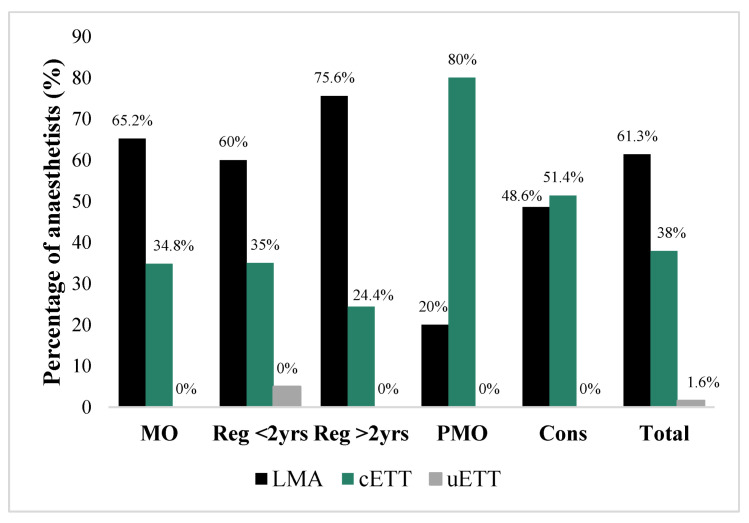
Choice of airway for forearm fracture MUA by grade of anaesthetist.

**Table 1 children-09-01416-t001:** Anaesthesia technique details for scenarios.

Components	Use (Circle)	Sequence Order (Circle)								
Site intravenous access	Y/N	N/A	1	2	3	4	5	6	7	8
Insert/suction naso/orogastric tube	Y/N	N/A	1	2	3	4	5	6	7	8
Pre-oxygenation	Y/N	N/A	1	2	3	4	5	6	7	8
Induction	Intravenous/Inhalational	N/A	1	2	3	4	5	6	7	8
Cricoid pressure	Y/N	N/A	1	2	3	4	5	6	7	8
Appropriate bag-mask-ventilation	Y/N	N/A	1	2	3	4	5	6	7	8
Muscle relaxant Suxamethonium (Sux) Rocuronium (Roc) Cis/-Atracurium (Cis-Atr) None (Ø)	Circle drug of choice Sux/Roc/Cis-Atr/Ø	N/A	1	2	3	4	5	6	7	8
Definitive airway control Laryngeal mask airway (LMA) Cuffed endotracheal tube (cETT) Uncuffed endotracheal tube (uETT)	LMA/cETT/uETT	N/A	1	2	3	4	5	6	7	8

**Table 2 children-09-01416-t002:** Responses according to grade of anaesthetist.

Grade of Anaesthetist	N	%
Medical Officer	23	18.3
Junior registrar (<2 years)	20	15.9
Senior registrar (>2 years)	41	32.5
Principal medical officer	5	3.9
Consultant	37	29.4
Total	126	100

**Table 3 children-09-01416-t003:** Anaesthesia Technique for Laparotomy for Suspected Small Bowel Obstruction.

	Insert/suction NG tube		PreO_2_		Cricoid		Induction		Paralysis				BMV		Airway			RSI Technique		
	No	Yes	No	Yes	No	Yes	IV	Gas	Sux	Roc	Cis	None	Yes	No	LMA	cETT	uETT	Classic	Controlled	Other
MO	5 (21.7)	18 (78.3)	3 (13)	20 (87)	11 (47.8)	12 (52.2)	20 (87)	3 (13)	8 (34.8)	10 (43.5)	1 (4.3)	4 (17.4)	7 (30.4)	16 (69.6)	0	21 (91.3)	2 (8.7)	13 (56.5)	5 (21.7)	5 (21.7)
Reg <2yrs	7 (35)	13 (65)	0	20 (100)	9 (45)	11 (55)	17 (85)	3 (15)	3 (15)	9 (45)	6 (30)	2 (10)	5 (25)	15 (75)	1 (5)	19 (95)	0	11 (55)	5 (25)	3 (15)
Reg >2yrs	7 (17.1)	34 (82.9)	1 (2.4)	40 (97.6)	20 (48.8)	21 (51.2)	40 (97.6)	1 (2.4)	14 (34.2)	23 (56.1)	4 (9.7)	0	18 (43.9)	23 (56.1)	0	41 (100)	0	19 (46.3)	17 (41.5)	4 (9.8)
PMO	2 (40)	3 (60)	0	5 (100)	2 (40)	3 (60)	5 (100)	0	0	2 (40)	2 (40)	1 (20)	1 (20)	4 (80)	0	5 (100)	0	3 (60)	0	2 (40)
Cons	8 (21.6)	29 (78.4)	0	37 (100)	13 (35.1)	24 (64.9)	35 (95.6)	0	19 (54.3)	10 (28.6)	4 (11.4)	2 (5.7)	13 (35.1)	24 (64.9)	0	35 (95.6)	0	21 (58.3)	13 (36.1)	2 (5.6)
Total	29 (23) n (%)	97 (77) n (%)	4 (3.2) n (%)	122 (96.8) n (%)	55 (43.7) n (%)	71 (56.3) n (%)	117 (94.3) n (%)	7 (5.7) n (%)	44 (35.5) n (%)	54 (43.6) n (%)	17 (28.6) n (%)	9 (7.3) n (%)	44 (34.9) n (%)	82 (65.1) n (%)	1 (2.4) n (%)	121 (97.6) n (%)	2 (1.6) n (%)	67 (53.6) n (%)	40 (32.5) n (%)	16 (12.9) n (%)

## Data Availability

All archived datasets may be accessed on request from the corresponding author, L.D.

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
