# Peer review of "A Survey of Paediatric Rapid Sequence Induction in a Department of Anaesthesia"

_children, 2022, doi:10.3390/children9091416_

Round 1

Reviewer 1 Report

This single institution survey reports a wide variety in practice of RSI in children and neonates. The findings are consistent with previous evidence.

The paper is easy to follow and read. There are only minor typos in the manuscript.

The authors may wish to elaborate of ways to reduce the variability in their practices especially that of MO (I assume these are not medically trained).

Author Response

Minor typos to be revised.

Conclusion will be updated to include ways in which to reduce variability in practice such as intra-departmental training and guidelines.

Thank you.

Reviewer 2 Report

This is an interesting descriptive study of RSI practice in one university anaesthesia department. However, it isn’t clear what the broader implications are. There is no clear evidence that “classic” is better or worse than “controlled” RSI, except that more experienced consultants are more likely to use controlled in neonates and younger infants. That suggests experiential knowledge supporting the technique, but does not really provide evidence of greater or lesser safety or efficacy. The authors recognize that significant morbidity from aspiration during anaesthetic induction is exceedingly rare in the paediatric and neonatal population, no matter which technique is used.

We don’t know the total size of the department, nor the distribution of emergency paediatric cases. It is impossible to interpret the preferences of different levels of anaesthetist without a description of the levels of training and degrees of responsibility. Medical officers and primary medical officers are not a role in other countries — what is their training? Who supervises registrars? Surely junior registrars are not anaesthetizing for emergency neonatal surgery unsupervised, in which case their attending would be making the decisions about technique, so their responses are not independent.

The data may provide information to guide an internal policy in that one department, or may prompt other departments to discuss the same issues, but it does not provide clear guidance as to a preferred approach. If the study prompted a departmental discussion and policy, and that was described, then the authors would be in a position to recommend this approach to other departments. 

Author Response

Thank you for the review.

Introduction will be improved to demonstrate that an the implementation of intra-departmental guidelines may improve practice variability, with less emphasis on controlled RSI improving safety and outcomes as this was not proved directly in this study.

MO and PMO clinical experience will be defined. Departmental size and response rate will be defined more clearly in methods and results sections.

Method will explain clearer that this is a survey of what the individual practitioner would do should they (hypothetically) be the sole individual choosing how the induction should happen. 

Conclusion will be changed to rather emphasize that these findings may be used to prompt internal training, guidelines, and policy that promote the benefits of controlled RSI and decrease practice variability between anaesthetists who have varying experience in paediatric patients.

Reviewer 3 Report

Dear colleagues,

I read your work with interest and appreciated the administration of clinical cases in the questionnaire and the discussion in which important topics are addressed such as the best pharmacological options (neuromuscular block), the choice of the entotracheal tube based on the actual anatomy of the pediatric airways , the need to encourage the use of the controlled RSI technique.

The authors present an interesting survey on pediatric rapid sequence induction in an anesthesia department.

Despite the limit of the work, represented by a study conducted on a single department, the results presented offer interesting food for thought on the subject, which is still controversial.

The results and, above all, the discussion correctly address important topics in pediatric airway management such as the different approaches of professionals based on the degree of pediatric experience gained, the need for more training on the topic studied and on the most suitable drugs, the still current use of cricoid pressure, the choice of the endotracheal tube on the basis of an anatomical conception of the airways perhaps to be updated.

I found very interesting the administration in the questionnaire of the clinical cases to be addressed, all appropriate.

It would be interesting to expand the survey to a much larger, international population of anesthetists.

The manuscript is clear and pleasant to read.

Bibliographic references are relevant and numerically adequate. There is only one self-citation and I think it can be tolerated.

The manuscript as a whole is scientifically valid and the topic is interesting.

The results of the manuscript are reproducible based on the details provided in the methods section.

The figures and tables are adequate.

The conclusions are consistent.

All ethical statements are present, including the authorization of the Ethics Committee.

Author Response

Dear Colleague,

Your review of our article is much valued. Thank you for your comments and for your time.

Round 2

Reviewer 2 Report

This version is considerably improved, and may now be of interest to anaesthetists who wish to institute department-wide reviews of practice and implement policies. It may also be of value to non-anaesthesiologist physicians doing emergency intubations (e.g. emergency or critical care paediatricians), encouraging them to think about their practice.